# Otoacoustic emissions but not behavioral measurements predict cochlear nerve frequency tuning in an avian vocal communication specialist

Diana M Karosas[1], Leslie Gonzales[2], Yingxuan Wang[1], Christopher Bergevin[3], Laurel H Carney[1,2], Kenneth S Henry[1,2,4]*

[1]Department of Biomedical Engineering, University of Rochester, Rochester, United States; [2]Department of Neuroscience, University of Rochester, Rochester, United States; [3]Department of Physics and Astronomy, York University, Toronto, Canada; [4]Department of Otolaryngology, University of Rochester, Rochester, United States

## eLife Assessment

In contrast with mammals, measures of cochlear tuning in budgerigars do not match the frequency dependence of behavioral tuning. Earlier behavioral data in the budgerigar had shown good selectivity at around 3–4 kHz, but it was unknown whether this unusual selectivity arose in the inner ear or was a more central adaptation. The authors measured both auditory-nerve tuning curves and stimulus-frequency otoacoustic emissions and found fairly normal-looking cochlear tuning in the budgerigar. These **important** findings imply that any behavioral/perceptual differences in frequency selectivity are likely more central in original. These **solid** new data also provide significant support for the utility of otoacoustic estimates of cochlear tuning.

*For correspondence:
kenneth_henry@urmc.rochester.edu

**Abstract** Frequency analysis by the cochlea forms a key foundation for all subsequent auditory processing. Stimulus-frequency otoacoustic emissions (SFOAEs) are a potentially powerful alternative to traditional behavioral experiments for estimating cochlear tuning without invasive testing, as is necessary in humans. Which methods accurately predict cochlear tuning remains controversial due to only a single animal study comparing SFOAE-based, behavioral, and cochlear frequency tuning in the same species. The budgerigar (*Melopsittacus undulatus*) is a parakeet species with human-like behavioral sensitivity to many sounds and the capacity to mimic speech. Intriguingly, previous studies of critical bands, psychophysical tuning curves, and critical ratios in budgerigars show that behavioral tuning sharpness increases dramatically with increasing frequency from 1 to 3.5 kHz, doubling once per octave with peak tuning sharpness from 3.5 to 4 kHz. The pattern contrasts with slower monotonic growth of behavioral tuning sharpness with increasing frequency in other animals, including most avian species, suggesting a possible auditory specialization in budgerigars. We measured SFOAE-based and cochlear-afferent tuning in budgerigars, for comparison to previously reported behavioral results. SFOAE-based and cochlear-afferent tuning sharpness both increased monotonically and relatively slowly for higher frequencies, in contrast to the behavioral pattern. SFOAE-based tuning in budgerigars accurately predicted cochlear frequency tuning, and both measures aligned with typical patterns of cochlear tuning in other species. Divergent behavioral tuning in budgerigars is unlikely attributable to the periphery and could reflect specializations for central processing of masked signals. Our findings highlight the value of SFOAEs for estimating cochlear tuning and caution against

direct inference of peripheral tuning from behavioral critical bands, psychophysical tuning curves, and critical ratios.

## Introduction

Cochlear frequency tuning provides the foundation for tonotopic processing by the auditory system (*Bourk et al., 1981*; *Schreiner and Langner, 1997*; *Song et al., 2024*). Otoacoustic emissions (*Kemp, 1978*) and behavioral testing (*Moore, 1978*) are two methods currently available to evaluate cochlear tuning in humans, and in other cases for which direct measurement is not feasible. Which methods accurately predict cochlear tuning remains controversial due to limited studies comparing otoacoustic, behavioral, and cochlear tuning in the same species.

Stimulus-frequency otoacoustic emissions (SFOAEs) are low-level acoustic signals recorded in response to tones that depend on active processes within the cochlea (*Kemp, 1978*; *Shera and Guinan, 1999*). SFOAE delay is expected to vary with cochlear tuning sharpness based on filter theory, with longer delays implicating sharper tuning. Previous studies found long SFOAE delays in humans, consistent with exceptional tuning sharpness (*Shera et al., 2002*). However, otoacoustic tuning estimation remains an indirect approach requiring validation. Behavioral experiments infer tuning from masking patterns, typically based on variation in tone thresholds across maskers differing in spectral content (*Moore, 1978*). However, behavioral tuning estimates depend on central processing of noisy signals in addition to cochlear tuning, and results vary across stimulus paradigms with debate as to which methods better approximate actual tuning (*Oxenham and Shera, 2003*; *Ruggero and Temchin, 2005*; *Leschke et al., 2022*). Whether otoacoustic or behavioral methods provide closer estimates of actual cochlear frequency tuning remains unknown.

Several reports in nonhuman species have tested the extent to which SFOAE delays predict tuning of auditory-nerve responses at the output of the cochlea (*Shera et al., 2010*; *Joris et al., 2011*; *Bergevin et al., 2015*). These studies found that auditory-nerve tuning quality exceeds $N_{SFOAE}$ (SFOAE delay in stimulus cycles) by a 'tuning ratio' that decreases from ~3 to 1 from apex to base across several species (*Shera et al., 2010*). To our knowledge, only a single study has compared SFOAE and behavioral tuning estimates to auditory-nerve recordings in the same species, ferret (*Sumner et al., 2018*). SFOAE-based, behavioral, and auditory-nerve tuning quality in ferret all increased monotonically for higher frequencies, in broad accordance with one another (*Sumner et al., 2018*). A stronger test of SFOAE vs. behavioral tuning estimates could be obtained by studying species with specialized behavioral and/or physiological frequency-resolving abilities.

The budgerigar (*Melopsittacus undulatus*) is a parakeet species with well-known auditory processing abilities from behavioral studies (*Dooling et al., 1989*; *Dooling et al., 2000*; *Ryals et al., 2013*; *Henry et al., 2016*; *Henry et al., 2017*; *Wong et al., 2019*; *Henry et al., 2024*). Intriguingly, previous behavioral studies of critical bands and psychophysical tuning curves in budgerigars report rapidly increasing behavioral tuning sharpness with increasing frequency to a distinct peak from 3.5 to 4 kHz, beyond which tuning sharpness declines dramatically (*Figure 1*; *Saunders et al., 1978*; *Saunders et al., 1979*; *Saunders and Pallone, 1980*; *Dooling et al., 2000*). The pattern is unusual compared to results from other species, including birds with highly similar audiograms, suggesting a possible specialization of the budgerigar's auditory system (*Okanoya and Dooling, 1987*). Budgerigars' average behavioral tuning $Q_{10}$ increases by 2.65 dB/octave from 1 to 3.5 kHz (i.e., nearly doubling for each octave increase; slope of $10 * \log_{10}[Q_{10}]$). In contrast, slopes closer to 1 dB/octave have typically been reported in humans (e.g., 0.9 dB/octave; *Oxenham and Shera, 2003*) and various animal behavioral studies (ferret: 1.25 dB/octave in *Sumner et al., 2018*; macaque: 0.55 dB/octave in *Burton et al., 2018*; European starling: 0.95 dB/octave in *Langemann et al., 1995*).

Further evidence of potentially specialized frequency tuning in budgerigars comes from comparative studies of behavioral critical ratios. The critical ratio is the threshold signal-to-noise ratio for tone detection in wideband noise, which scales linearly with tuning bandwidth in the power spectrum model of auditory masking (*Fletcher, 1940*). In birds, most species have similar critical ratios to one another, increasing monotonically with higher frequencies by 2–3 dB/octave, as in humans and other mammals (*Figure 1b*; *Dooling et al., 2000*). In contrast, budgerigar critical ratios diverge markedly at mid-to-high frequencies, with ~8 dB lower (more sensitive) thresholds than other bird species from 3 to 4 kHz and intriguingly little variation in critical ratios below 4 kHz (*Dooling and Saunders, 1975*;

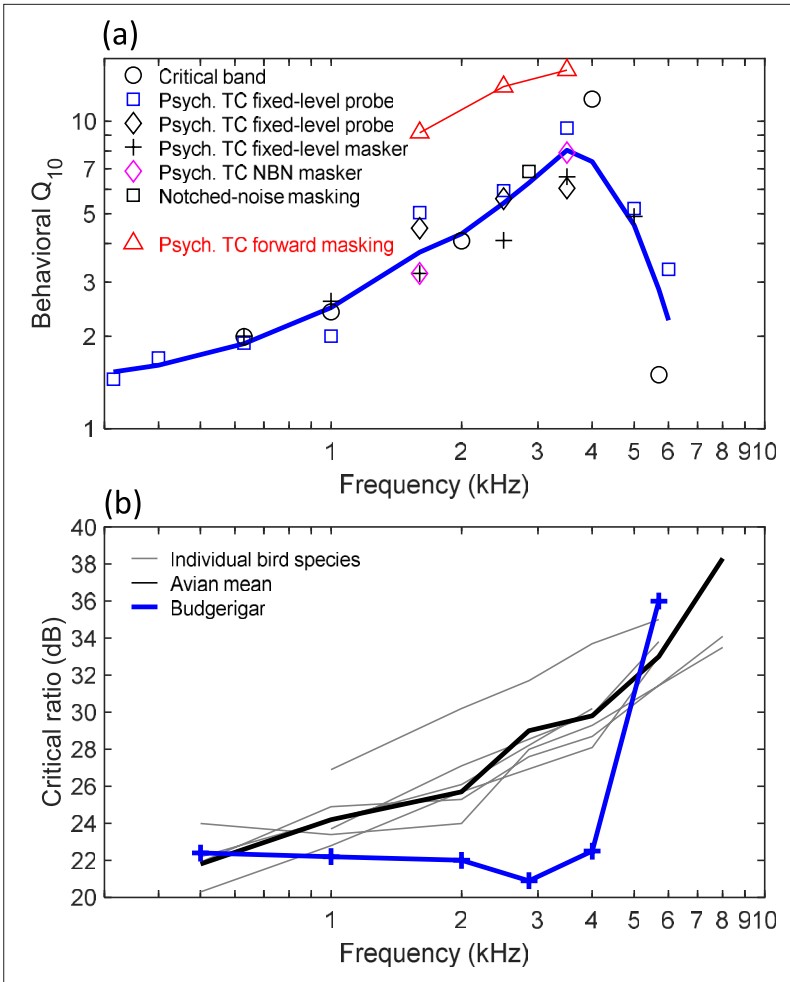

**Figure 1.** Behavioral frequency tuning in the budgerigar and comparison to other avian species. (**a**) Published behavioral studies employing multiple stimulus paradigms indicate rapidly increasing behavioral tuning sharpness to a distinct peak from 3.5 to 4 kHz. Tuning sharpness nearly doubles for each octave increase in frequency from 1 to 3.5 kHz (2.65 dB/octave). Red triangles are based on forward masking (**Kuhn and Saunders, 1980**); all other symbols indicate simultaneous masking results (black circles: **Saunders et al., 1978**; blue squares: **Saunders et al., 1979**; black diamonds: **Kuhn and Saunders, 1980**; black crosses: **Saunders and Pallone, 1980**; magenta diamonds: **Saunders et al., 1979**; black square: **Dooling et al., 2000**). The thick solid blue line is a LOESS fit to the simultaneous-masking results with a smoothing parameter of 0.65. TC: tuning curve; NBN: narrowband noise. (**b**) Behavioral critical ratios of the budgerigar are relatively constant to 4 kHz, in contrast to other species showing 2–3 dB/octave increasing critical ratios for higher frequencies. Thin gray lines: individual bird species tested by **Okanoya and Dooling, 1987** (cockatiel, *Nymphicus hollandicus*; canary, *Serinus canarius*; song sparrow, *Melospiza melodia*; swamp sparrow, *Melospiza georgiana*; starling, *Sturnus vulgaris*; zebra finch, *Taeniopygia guttata*); thick black line: avian mean of 10 species from **Dooling et al., 2000** (pigeon, *Columba livia*; red-winged blackbird, *Agelaius phoeniceus*; brown-headed cowbird, *Molothrus ater*; firefinch, *Lagonosticta segala*; also species from **Okanoya and Dooling, 1987**); thick blue line: budgerigar mean from **Dooling et al., 2000**.

*Okanoya and Dooling, 1987*; *Farabaugh et al., 1998*). Constant critical ratios indicate constant bandwidth in the power spectrum model, and consequently, increasing tuning quality by 3 dB/octave, in general agreement with budgerigars' behavioral critical bands and psychophysical tuning curves. Notably, divergent critical ratios in the budgerigar occur despite similar audiogram shape in budgerigars to other avian species (*Okanoya and Dooling, 1987*; *Dooling et al., 2000*). Thus, unusual behavioral tuning in budgerigars is not simply explainable by declining audibility above 4 kHz. On the other hand, the impact of sloping audibility on behavioral tuning estimates remains poorly understood.

Because auditory-nerve tuning has not been measured in budgerigars, it remains unknown whether budgerigars' unusual critical bands, psychophysical tuning curves, and critical ratios reflect

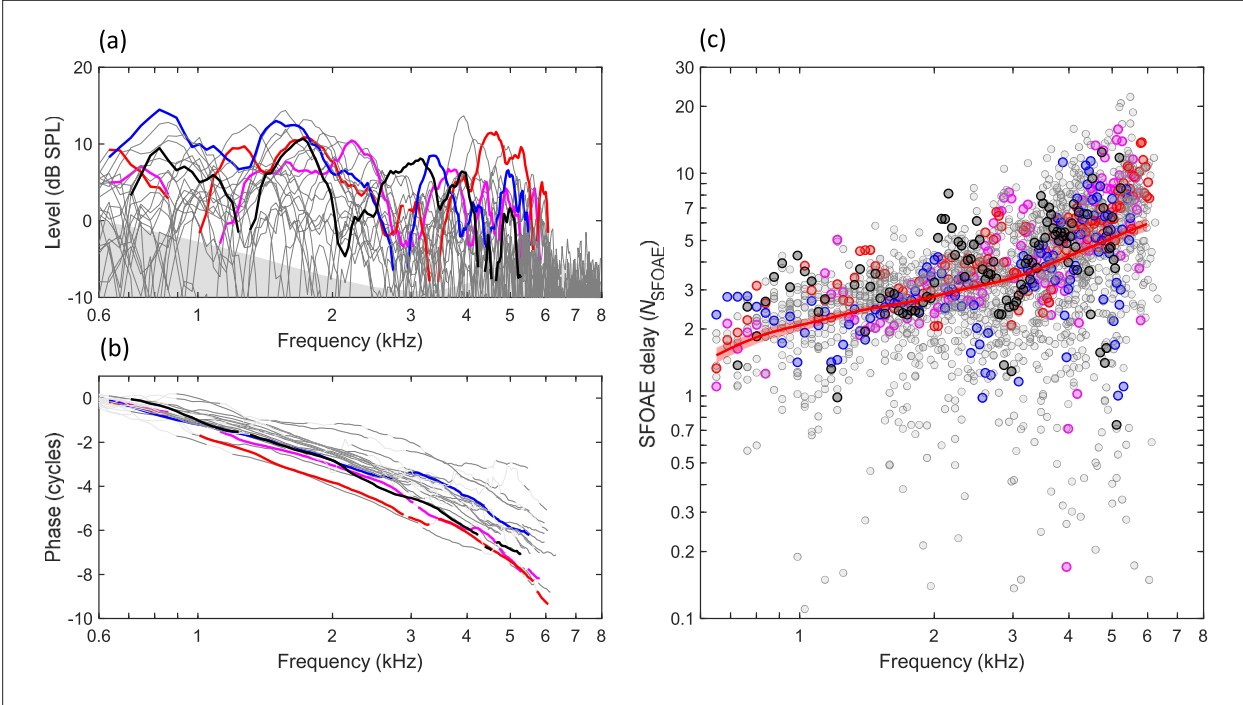

**Figure 2.** Budgerigar stimulus-frequency otoacoustic emissions (SFOAEs) of 22 ears in 14 animals. (**a**) SFOAE level ranged from 5 to 15 dB at stimulus frequencies from 0.6 to 6 kHz, showing multiple spectral notches within this frequency range, and descended into the noise floor (gray shaded region) at higher frequencies. Thick colored lines highlight SFOAE level in four representative ears. (**b**) SFOAE phase decreased monotonically for higher stimulus frequencies. (**c**) SFOAE delay in stimulus cycles, $N_{SFOAE}$, calculated from individual phase gradients in (**b**). The thick red line and shaded region show the weighted mean and 95% CI, respectively (Gaussian weighting function; sigma: 0.25 octaves; 1000 bootstrap repetitions; see also *Figures 3 and 5* for illustration of $N_{SFOAE}$ CIs). Data from the same representative ear are presented in the same color across panels.

specialized cochlear tuning, and if so, whether the pattern is detectable using SFOAEs. The present study compared the extent to which SFOAE and existing behavioral tuning measures predict actual cochlear tuning in budgerigars, a species that does not show slow monotonic growth of behavioral tuning sharpness with increasing frequency (*Figure 1*). SFOAE-based tuning, estimated with low-level swept-tone stimuli, was compared to measures of auditory-nerve tuning and to previously reported behavioral measures of frequency tuning. In budgerigars, SFOAE and auditory-nerve tuning sharpness both increased relatively slowly and monotonically with frequency, as in most other species, and in contrast to previously reported behavioral results. Thus, SFOAE delay predicted auditory-nerve tuning sharpness with greater accuracy than behavioral critical bands, psychophysical tuning curves, and critical ratios, and we find no evidence of a peripheral specialization in budgerigars.

## Results
### Budgerigar SFOAEs
SFOAEs were recorded from 22 ears in 14 animals using swept-frequency tones presented at 40 dB SPL. SFOAE level (*Figure 2a*) typically ranged from 5 to 15 dB at stimulus frequencies below 5–6 kHz and descended into the noise floor at higher frequencies. Within the frequency of measurable emissions, SFOAEs in individual ears (*Figure 2a*, colored lines) typically showed three or more deep spectral notches in emission level. Frequency regions near spectral notches, or for which SFOAE level was within 10 dB of the estimated noise floor, were excluded from subsequent analyses as described below (see 'Materials and methods: SFOAE processing'). SFOAE phase decreased monotonically with increasing frequency and was relatively consistent across individual ears (*Figure 2b*). Finally, SFOAE delay was estimated from the phase gradient of the response in stimulus cycles, that is, as $N_{SFOAE}$ (*Figure 2c*). So calculated, $N_{SFOAE}$ increased monotonically with increasing frequency, consistent with gradually increasing tuning quality along the length of the cochlea from apex to base. The local mean

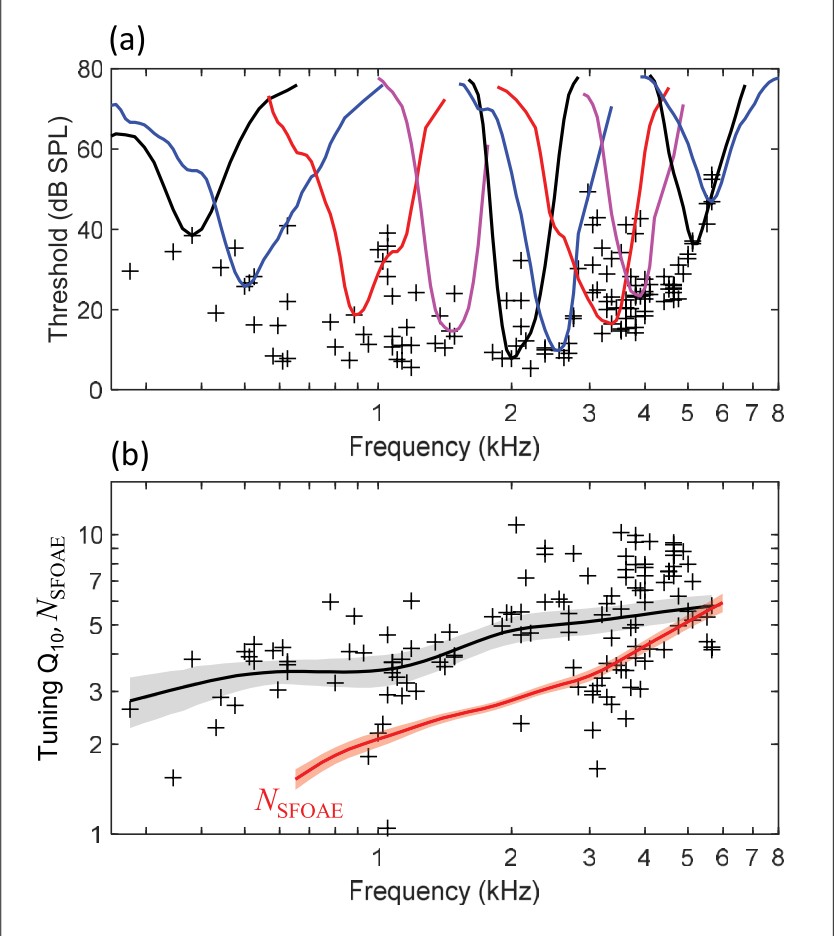

**Figure 3.** Budgerigar auditory-nerve recordings from 127 fibers in 6 animals. (**a**) Representative tuning curves (colored lines) show the threshold for excitation of the neural response as a function of tone frequency. Tuning curves are V-shaped and are approximately symmetrical around characteristic frequency (CF) on a log-frequency axis. Black crosses show the threshold at the CF for all recorded fibers. (**b**) Tuning-curve $Q_{10}$ increases for higher CFs across the neural population. The black trend line and gray shaded region show the weighted mean $Q_{10}$ (Gaussian weighting function; sigma: 0.5 octaves) and 95% CI (1000 bootstrap repetitions), respectively. The thick red line shows $N_{SFOAE}$ replotted from **Figure 2**.

trendline for $N_{SFOAE}$ (**Figure 2c**, thick red line) was estimated using a Gaussian weighting with sigma of 0.25 octaves, fit to group delay (i.e., $N_{SFOAE} \times$ frequency) because this quantity was approximately normally distributed. 95% CIs in **Figure 2c** and throughout are based on 1000 bootstrap repetitions.

Linear mixed-effects modeling revealed a significant effect of frequency on SFOAE level ($F_{4,84} = 18.91$, $p<0.0001$), associated with slightly lower SFOAE level in the higher frequency bands (see Materials and methods: Statistical analysis). Moreover, a linear mixed-model analysis of $N_{SFOAE}$ revealed a significant effect of frequency ($F_{4,84} = 44.05$, $p<0.0001$) due to greater $N_{SFOAE}$ in higher frequency bands. The mean rate of increase in $N_{SFOAE}$ was 1.65 dB/octave. In summary, SFOAE results pooled across ears were consistent with monotonically increasing tuning sharpness from the cochlear apex to base.

## Auditory-nerve tuning curves

Auditory-nerve tuning curves (n=127) in budgerigars had CFs ranging from 0.28 to 5.65 kHz (median: 2.97 kHz; interquartile range: 1.13–3.88 kHz) and associated thresholds at CF as low as 5 dB SPL (**Figure 3a**, crosses). Individual tuning curves (**Figure 3a**, colored lines) appeared symmetric around CF on a log-frequency axis, with no clear indication of a secondary tail region of low-frequency sensitivity for stimulus levels up to 80 dB SPL (i.e., the highest level tested). $Q_{10}$, the measure of tuning-curve

sharpness, increased monotonically and significantly for higher CFs (*Figure 3b*; $t_{125}$=6.542, p<0.0001; *t*-test of linear regression slope between log-transformed variables) as in other animal species, without an apparent peak at 3.5–4 kHz. The local mean trendline for $Q_{10}$ (*Figure 3b*, thick black line) was calculated using a Gaussian weighting function with sigma of 0.5 octaves. The rate of increase for $Q_{10}$ with increasing frequency was 0.87 dB/octave, that is, lower than the 1.65 dB/octave rate observed for $N_{SFOAE}$ (*Figure 3b*; red line) and considerably lower than the 2.65 dB/octave rate for the behavioral literature (*Figure 1*; see below for further comparison). $Q_{ERB}$ also increased monotonically for higher CFs (see below) and exceeded $Q_{10}$ by a median factor of 1.76 (interquartile range: 1.65–1.89).

## Defining the tuning ratio, *r*, for SFOAE-based $Q_{ERB}$ prediction

Previous studies (e.g., *Shera et al., 2002*; *Shera et al., 2010*) established the concept of a tuning ratio, *r*, assumed to vary slowly with frequency and fit to the empirically derived quotient of auditory-nerve $Q_{ERB}$ and $N_{SFOAE}$, as indicated in *Equation 1*.

$$r = \frac{Q_{ERB}}{N_{SFOAE}} \tag{1}$$

The value of *r* in mammals decreases from ~3 at the apical end of the cochlea to 1 for frequencies processed more basally (*Shera et al., 2010*). *r* appears largely conserved across related species, such that multiplying $N_{SFOAE}$ by *r* from a different species or species group accurately predicts auditory-nerve $Q_{ERB}$ in many cases (*Shera et al., 2010*; *Joris et al., 2011*; *Sumner et al., 2018*). Given substantial differences in cochlear morphology between birds and mammals (*Gleich and Manley, 2000*), we calculated *r* from chicken (*Gallus gallus domesticus*) to predict $Q_{ERB}$ from $N_{SFOAE}$ results in the budgerigar. Chicken data were selected rather than barn owl (*Tyto alba*; *Bergevin et al., 2015*) due to greater similarity of the chicken's cochlear morphology (*Gleich and Manley, 2000*) and frequency limits of hearing (chicken: 9.1–7,200 Hz [*Hill et al., 2014*]; budgerigar: 77–7,600 Hz [*Heffner et al., 2016*]; frequency ranges for which behavioral audiometric thresholds are less than 60 dB SPL). No attempt was made to adjust *r* for a possible species difference in the apical-basal transition (*Shera et al., 2010*; *Shera and Charaziak, 2019*) due to similar hearing ranges between chicken and budgerigar as well as uncertainty as to how this concept from the mammalian SFOAE literature might apply to the substantially shorter avian cochlea.

Auditory-nerve $Q_{10}$ from two previous chicken studies (*Manley et al., 1991*; *Saunders et al., 1996*) increases monotonically for higher frequencies (*Figure 4a*). $N_{SFOAE}$ in chicken also increases monotonically for higher frequencies (*Bergevin et al., 2008*; *Figure 4a*, red line), similar to the pattern observed for auditory-nerve $Q_{10}$. Finally, the $Q_{10}$ trendline was fit to pooled data pooled across the two auditory-nerve studies and multiplied by 1.76 (approximating auditory-nerve $Q_{ERB}$; see above) and divided by $N_{SFOAE}$ to yield chicken *r* (*Figure 4b*). Chicken *r* decreased from a maximum value of ~7 at 500 Hz to 3–4 at frequencies from 1 to 3.35 kHz.

Note that empirical evaluation of chicken *r* was limited to frequencies less than 3.35 kHz due to the scala tympani approach of the two auditory-nerve studies, which limited sampling from high-CF fibers (*Manley et al., 1991*; *Saunders et al., 1996*). We therefore extended *r* to 4.6 kHz by first extrapolating chicken $Q_{10}$ to 4.6 kHz. Extrapolation was performed with a linear regression model fit to $Q_{10}$ data from fibers with CFs greater than 1.7 kHz (*Figure 4a*). Extrapolated $Q_{10}$ (predicted means ±1 SE; *Figure 4a*) trended slightly upward in frequency in accordance with the regression model slope. Extended-frequency *r* (*Figure 4b*, magenta lines from 3 to 4.6 kHz), calculated as the quotient of extrapolated $Q_{10}$ over $N_{SFOAE}$, trended downward with increasing frequency from 3.35 to 4.6 kHz.

Finally, budgerigar *r* was calculated using the same empirical method, but without extrapolation due to broad sampling of CFs with our auditory-nerve approach. Budgerigar *r* was marginally lower than that of chickens and followed the same trajectory with increasing frequency, decreasing from a maximum of ~4 at 700 Hz to ~2 at 5.5 kHz (*Figure 4b*, thick black line).

## SFOAEs but not behavioral measurements accurately predict auditory-nerve frequency tuning in the budgerigar

SFOAE-based predictions of cochlear tuning in budgerigars were made by multiplying $N_{SFOAE}$ by chicken *r* as a function of frequency (*Figure 5*; magenta). Budgerigar SFOAE predictions, plotted together with auditory-nerve $Q_{ERB}$ (thick black line) and average behavioral tuning sharpness in this

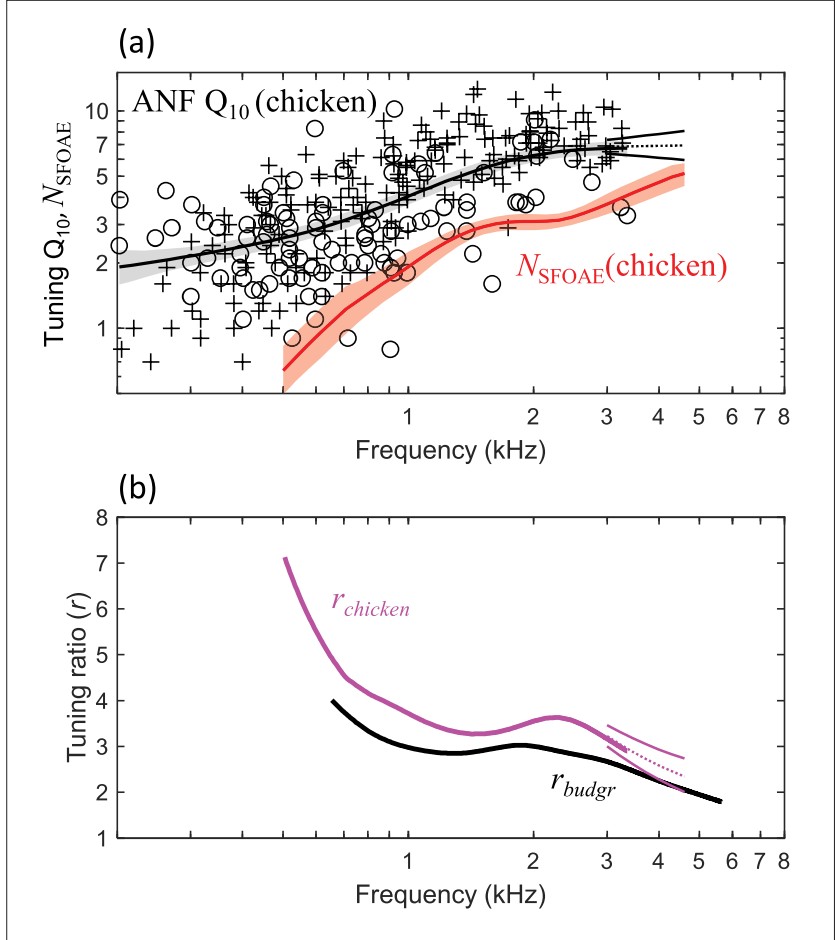

**Figure 4.** Estimation of the stimulus-frequency otoacoustic emission (SFOAE) tuning ratio, *r*, from published chicken studies. (**a**) $Q_{10}$ values of chicken auditory-nerve fibers (ANFs; n=289; crosses: *Saunders et al., 1996*; circles: *Manley et al., 1991*) increase monotonically for higher characteristic frequencies (CFs), similar to the rate at which chicken $N_{SFOAE}$ (*Bergevin et al., 2008*; n=9 animals) increases with frequency. Thick lines and shaded regions show weighted means (Gaussian weighting function; sigma: 0.5 octaves) and 95% CIs (1000 bootstrap repetitions), respectively; back lines extending from 3 to 4.6 kHz show extrapolated $Q_{10}$ (dotted: mean; solid ±1 SE; see text). (**b**) The tuning ratio, *r*, of chicken (magenta) compared to that of budgerigars (black). *r* is the quotient of auditory-nerve $Q_{ERB}$ over $N_{SFOAE}$ at the same frequency. Magenta lines extending form 3–4.6 kHz show extended-frequency *r* of chicken based on extrapolated $Q_{10}$. *r* is marginally higher in the chicken than budgerigar, and trends downward with increasing frequency in both species.

species based on critical bands and psychophysical tuning curves (thick blue line), highlight the similarities and differences across these measures. Behavioral $Q_{10}$ values from the literature (*Figure 1*) were scaled by a factor of 1.76 to approximate $Q_{ERB}$. Whereas $N_{SFOAE}$ predictions made using both the empirical and extended-frequency forms of chicken *r* reasonably approximated budgerigar auditory-nerve $Q_{ERB}$ measurements, running parallel to and slightly above the mean trendline, the behavioral results show a fundamentally different pattern characterized by a noticeably greater slope from 0.5 to 3.5 kHz, distinct peak from 3.5 to 4 kHz, and steep roll off above 4 kHz. Thus, SFOAE results provided closer estimates of actual auditory-nerve $Q_{ERB}$ than did behaviorally estimated tuning measures in the budgerigar, and we find no evidence that unusual behavioral tuning in budgerigars arises peripherally.

## Discussion

The present study measured SFOAEs in budgerigars to predict cochlear frequency tuning. Tuning curves were also recorded from budgerigar single auditory-nerve fibers for direct comparison. SFOAE level remained above the noise floor for frequencies below 6 kHz and trended downward with

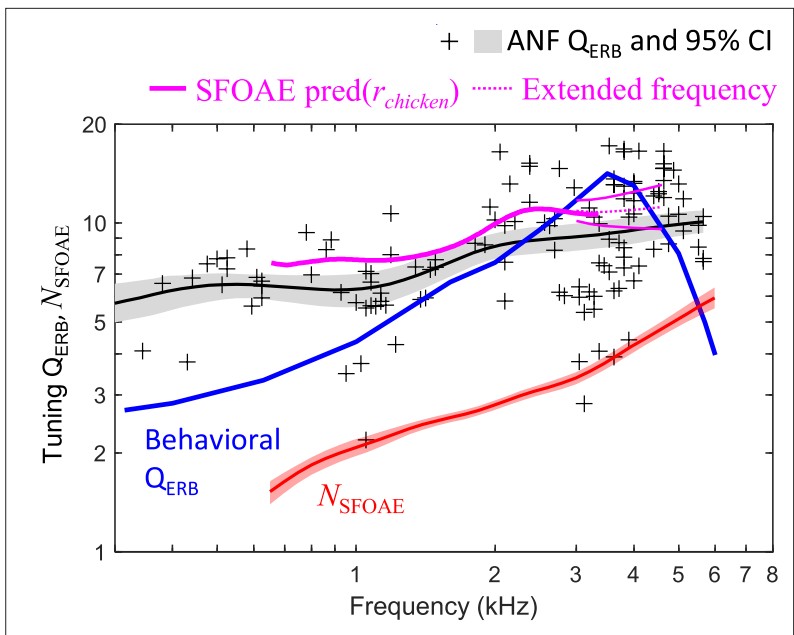

**Figure 5.** Comparison of stimulus-frequency otoacoustic emission (SFOAE) predictions, behavioral frequency tuning, and directly measured auditory-nerve $Q_{ERB}$ in the budgerigar (n=127 fibers from six animals). The SFOAE prediction based on $r$ from chicken (empirical range: thick magenta line; extended-frequency range [means ± 1SE]: thin magenta lines) slightly exceeds the upper 95% confidence limit for mean auditory-nerve $Q_{ERB}$ (Gaussian weighting function; sigma: 0.5 octaves; 1000 bootstrap repetitions) and shows the same frequency dependence. In contrast, behavioral $Q_{ERB}$ (thick blue line) shows a fundamentally different profile characterized by a peak from 3.5 to 4 kHz. Mean $N_{SFOAE}$ of budgerigars (thick red line) is replotted from **Figure 2**.

increasing frequency. SFOAE delay expressed as $N_{SFOAE}$ increased steadily for higher stimulus frequencies, consistent with increasing tuning quality along the length of the cochlea in the 650 Hz to 6 kHz range. Moreover, quantitative SFOAE predictions calculated as the product of budgerigar $N_{SFOAE}$ and chicken $r$ showed close agreement with measured auditory-nerve $Q_{ERB}$ across the frequency range available for comparison. In contrast, previous behavioral studies of budgerigar psychophysical tuning curves, critical bands, and iso-level masking functions highlight rapidly increasing tuning quality from 1 to 3.5 kHz and a prominent peak in tuning sharpness from 3.5 to 4 kHz, suggestive of an auditory specialization compared to other avian species (**Dooling and Saunders, 1975**; **Saunders et al., 1978**; **Saunders et al., 1979**; **Saunders and Pallone, 1980**). Overall, the results show that SFOAE delays in budgerigars better predict auditory-nerve tuning sharpness compared to previously reported behavioral frequency-tuning estimates based on masking, and that unusual behavioral tuning in budgerigars is unlikely to arise in the cochlea.

SFOAE delay expressed as $N_{SFOAE}$ increases for higher frequencies in a wide range of taxa, in qualitative agreement with patterns of auditory-nerve tuning quality in these species (macaque monkeys: **Joris et al., 2011**; chinchilla: **Shera et al., 2010**; gecko: **Bergevin and Shera, 2010**; cat and guinea pig: **Shera et al., 2002**; ferret: **Sumner et al., 2018**; chicken: **Bergevin et al., 2008**; barn owl: **Bergevin et al., 2015**). More detailed quantitative prediction of $Q_{ERB}$ requires scaling $N_{SFOAE}$ by $r$, where $r$ is the empirically defined quotient of $Q_{ERB}$ over $N_{SFOAE}$. In the budgerigar, use of $r$ from chicken predicted auditory-nerve $Q_{ERB}$ with good quantitative precision across the full range of frequencies evaluated (**Figure 5**). The precision of SFOAE predictions is attributable to similarity of $r$ between the budgerigar and chicken (**Figure 4b**). The value of $r$ in budgerigars decreased from ~4 at 700 Hz to ~2 at 5.5 kHz and was marginally higher in the chicken with similar frequency dependence. While the selection of chicken $r$ for our study was reasonable, given similar frequency limits of hearing and cochlear anatomy between chicken and budgerigar (**Gleich and Manley, 2000**; **Heffner et al., 2016**; **Hill et al., 2014**), note that other species show lower $r$ that, if used for SFOAE predictions in the budgerigar, would considerably underestimate auditory-nerve $Q_{ERB}$. The value of $r$ in mammals decreases from ~3 to 1 from apex to base (**Shera et al., 2010**), whereas $r$ in the barn owl ranges from 1 to 2 (**Bergevin**

*et al., 2015*). The reasons for species differences in *r* are unknown, though level dependence of *r* in previous studies suggests that differences in hearing sensitivity could be one factor (*Bergevin et al., 2015*). Moreover, mammalian studies suggest an apical-basal transition point for *r*, basal to which *r* approaches ~1 and apical to which *r* varies inversely with frequency (*Shera et al., 2010*; *Shera and Charaziak, 2019*). Variation in the location of the apical-basal transition could also contribute to species differences in *r*. Whether an apical-basal transition occurs in the avian cochlea is unknown, but it is intriguing to note that *r* from the budgerigar and chicken shows some resemblance to the apical portion of *r* in cat, guinea pig, chinchilla, and probably other mammalian species (*Shera et al., 2010*; *Shera and Charaziak, 2019*). Because *r* is such a key parameter in otoacoustic estimation of cochlear frequency tuning, further studies of its taxonomic variation, level dependence, and physiological basis are highly warranted (*Bergevin et al., 2008*; *Manley, 2022*).

Measurements of chicken auditory-nerve tuning were drawn from two previously published studies and were limited to CFs less than 3.35 kHz due to their recording methodology (*Manley et al., 1991*; *Saunders et al., 1996*). CFs likely exist up to at least 5 kHz based on the chicken's behavioral audiogram (*Hill et al., 2014*), but were not sampled due to the scala tympani approach employed by these studies (*Manley et al., 1991*; *Saunders et al., 1996*). Consequently, computing *r* across the full range of the chicken $N_{SFOAE}$ function (0.5–4.6 kHz) required moderate extrapolation of the auditory-nerve $Q_{10}$ trendline from 3.4 to 4.6 kHz based on lower CF (1.7–3.35 kHz) data. While probably reasonable given continued increase of $Q_{10}$ above ~3 kHz in many other avian species (*Sachs et al., 1974*; *Gleich and Manley, 2000*), reliance in part on extrapolated chicken $Q_{10}$ remains a limitation of this study. It should also be noted that $Q_{10}$ data from a third chicken study (*Salvi et al., 1992*) were considered but ultimately not included in our estimation of chicken *r* because CFs were less than 2 kHz and the reported $Q_{10}$ values appeared somewhat discrepant (i.e., some reported values as high as 24).

The estimates of behavioral frequency selectivity referenced in the present study were originally obtained using three simultaneous masking paradigms: psychophysical tuning curves for probe-tone detection in tonal or narrowband maskers, critical bands for tone detection in bandlimited noise, and critical ratios for tone detection in wideband noise (*Dooling and Saunders, 1975*; *Saunders et al., 1978*; *Saunders et al., 1979*; *Saunders and Pallone, 1980*; *Kuhn and Saunders, 1980*; *Dooling et al., 2000*). All three behavioral measures show that perceptual frequency selectivity in budgerigars is greatest from 3.5 to 4 kHz and declines markedly at higher and lower frequencies. Suggestive of a possible auditory specialization, this pattern differs substantially from behavioral findings in humans and other model species, which show monotonically increasing behavioral frequency selectivity for higher frequencies. Critically, the behavioral findings contrast with physiological tuning estimates of the current study, which show monotonically increasing SFOAE phase gradients with increasing frequency as well as monotonically increasing auditory-nerve $Q_{10}/Q_{ERB}$ values for higher CFs. Thus, neither SFOAEs nor auditory-nerve recordings suggest the existence of a peripheral auditory specialization that could explain prior behavioral findings in this species.

While the reason for disparate behavioral and SFOAE/auditory-nerve findings in budgerigars is unknown, the mismatch highlights the need for caution in inferring cochlear frequency selectivity using behavioral critical bands, psychophysical tuning curves, critical ratios, and simultaneous masking procedures. While the budgerigar behavioral studies of frequency tuning were conducted several decades ago, the methods are based on the still widely used power-spectrum model of auditory masking and remain commonly employed in animal behavioral research (e.g., critical bands and ratios: *Yost and Shofner, 2009*; *King et al., 2015*; simultaneous masking: *Burton et al., 2018*). Forward-masking procedures are hypothesized to more accurately predict cochlear frequency tuning in humans (*Shera et al., 2002*; *Joris et al., 2011*; *Sumner et al., 2018*), but evidence from nonhuman studies is mixed. *Sumner et al., 2018* found that forward-masking behavioral results better predicted auditory-nerve tuning than simultaneous masking in the ferret (*Sumner et al., 2018*). On the other hand, *Ruggero and Temchin, 2005* argued that forward-masking behavioral paradigms can substantially overestimate cochlear frequency selectivity, highlighting several cases where simultaneous-masking procedures produce close quantitative predictions of auditory-nerve tuning (e.g., guinea pig, chinchilla, macaque; *Ruggero and Temchin, 2005*; see *Joris et al., 2011* for macaque auditory-nerve tuning). Interestingly in the budgerigar, psychophysical tuning curves obtained with pure-tone forward masking *Kuhn and Saunders, 1980* have $Q_{10}$ values that are greater than twice simultaneous-masking values at the same frequencies (*Figure 1*; red triangles), while also far exceeding (by two- or threefold)

the maximum auditory-nerve $Q_{10}$ values reported in *Figure 3*. It remains untested whether notched-noise forward masking procedures would produce closer estimates of auditory-nerve frequency tuning. In summary, at least in budgerigars, it appears unlikely that any of the various previously employed forward or simultaneous masking paradigms reasonably estimate cochlear frequency selectivity.

A potential explanation for the disparity between behavioral and the other measures of frequency tuning lies in the nature of the measurements. Otoacoustic and auditory-nerve responses both reflect peripheral physiological processes originating in the cochlea. However, behavioral measures of frequency selectivity necessarily involve the entire auditory pathway and central nervous system, including neural mechanisms for processing masked signals, and are best explained by models that include both peripheral and central auditory system processes (*Maxwell et al., 2020*; *Brennan et al., 2023*). The unique pattern observed in the budgerigar's behavioral $Q_{10}$ trendline may then reflect specializations for masked processing in the central auditory pathway of this highly vocal and gregarious species. While a previous study investigated neural tone-in-noise processing in the budgerigar's inferior colliculus, the study did not explore the impact of noise bandwidth on neural detection thresholds (*Wang et al., 2021*; which would be needed to estimate neural critical bands). Note that the frequency selectivity of inferior colliculus response maps obtained with single tones of varying frequency and level showed no obvious peak in tuning quality for neural CFs from 3.5 to 4 kHz (*Henry et al., 2016*; *Henry et al., 2017*), similar to the present auditory-nerve results. Finally, several histological studies of budgerigar cochlear morphology also revealed no obvious specializations consistent with a peripheral auditory specialization (*Manley et al., 1993*; *Wang et al., 2023*).

In conclusion, SFOAE-based and auditory-nerve measures of tuning sharpness increased for higher frequencies in the budgerigars, and quantitative SFOAE predictions based on *r* from a different bird species (chicken) closely approximated actual budgerigar auditory-nerve tuning. The pattern is similar to findings in other avian and mammalian species and contrasts with the unusual pattern of behavioral frequency tuning in budgerigars, which has maximum tuning quality from 3.5 to 4 kHz consistent with an auditory specialization. These results highlight the need for caution in interpreting behavioral measures of frequency tuning, which can show substantial deviation from cochlear tuning curves due to central processing mechanisms and other factors that necessarily influence behavioral detection of masked signals.

## Materials and methods

### Animals

Budgerigars were sourced from a local breeder or our institutional breeding program. SFOAEs were recorded from 22 ears in 14 budgerigars (7 female). Birds were 6–32 months old with a median age of 8.5 months at the time of SFOAE experiments. Auditory-nerve responses were recorded from six ears in six budgerigars (four females). Birds were 6–22 months old at the time of the auditory-nerve recordings (median: 7.5 months). All experiments were approved by the University of Rochester Committee on Animal Resources (UCAR-2011-015).

### SFOAE recordings

Animals were anesthetized for SFOAE recordings by subcutaneous injection of 0.08–0.12 mg/kg dexmedetomidine and 3–5.8 mg/kg ketamine. Following anesthesia, birds were placed in a stereotaxic apparatus located inside a double-walled acoustic isolation booth (Industrial Acoustics; 2 × 2.1 × 2.2 m). Temperature was maintained at 40°C throughout the experiment using a feedback-controlled heating pad (Harvard Apparatus Model 50-753, Edenbridge, KY, USA). Careful control of temperature was found to be critical for robust emission measurements. Breathing rate was monitored using a thermistor-based sensor.

After birds were placed in the stereotactic apparatus, a probe assembly consisting of a low-noise microphone (Etymotic ER10-B+, Elk Grove Village, IL, USA) and two earphones (Etymotic ER2, Elk Grove Village) was sealed to the ear using ointment. Acoustic stimuli were generated in MATLAB (The MathWorks, Natick, MA, USA; 50 kHz sampling frequency) and processed with a digital pre-emphasis filter (5000-point FIR) that corrected for the frequency response (i.e., magnitude and phase) of the system. The frequency response of the system was measured with the probe microphone at 249 linearly spaced frequencies (1 V peak amplitude) from 0.05 to 15.1 kHz. Stimuli were converted to

analog at maximum scale (10 V peak amplitude; National Instruments PCIe-6251, Austin, TX, USA), attenuated to the required levels (Tucker Davis Technologies PA5, Alachua, FL, USA), and amplified using a headphone buffer (Tucker Davis Technologies HB7).

Stimuli for measuring SFOAEs were swept tones, as in *Kalluri and Shera, 2013*. Two distinct acoustic signals were presented: a probe and a suppressor. The probe consisted of a tone linearly swept at a constant rate from 500 Hz to 8 kHz over 2.05 s, presented at 40 dB SPL. The suppressor consisted of a tone swept at a constant rate from 540 Hz to 8.04 kHz over 2.05 s, presented at 55 dB SPL. Stimuli were presented in repeating sequences in which the probe was presented first, then the suppressor, and finally the probe and the suppressor simultaneously. All stimuli had 25 ms raised-cosine onset and offset ramps. Each experiment consisted of 40 repetitions of the stimulus sequence. There was a half-second silent time gap between presentations of each stimulus sequence. Response waveforms were amplified by 40 dB (Etymotic ER10B+) prior to sampling at 50 kHz and storage on a computer hard drive.

After the completion of the experiments, birds were given a subcutaneous injection of 0.5 mg/kg atipamezole and placed in a heated recovery chamber until fully alert.

## Auditory-nerve recordings

The surgical procedures for neurophysiological recordings in budgerigars have been described previously (*Wang et al., 2021*). Equipment is described above in the section on SFOAE recordings, except where noted. Briefly, animals were anesthetized with a weight-dependent subcutaneous injection of ketamine and dexmedetomidine, as indicated above, and a head-post was mounted to the dorsal surface of the skull with dental cement to facilitate head positioning. Animals were placed in a stereotaxic device inside a sound-attenuating chamber, with the beak projected downward. The right ear was aligned with an earphone/microphone assembly and sealed with silicone grease. Body temperature was maintained at 40°C using a homeothermic control unit, and breathing rate was monitored throughout experiments. Animals were maintained in an areflexic state through continuous subcutaneous infusion of additional ketamine (3–10 mg/kg/h) and dexmedetomidine (0.08–0.2 mg/kg/h), along with physiological saline, using a syringe pump. DPOAEs were recorded periodically throughout experiments to test for possible deterioration of inner-ear function (*Wong et al., 2019*). Animals were euthanized at the conclusion of recordings, typically after 10–12 h.

Access to the auditory nerve was accomplished by aspirating the right third of the cerebellum to reveal stereotaxic landmarks associated with the anterior semicircular canal. Recordings were made using pulled glass microelectrodes with 10–80 MΩ impedance. The tip of the electrode was positioned just anterior to a bony spur located on the medial surface of the ampulla of the anterior canal, and advanced into the brainstem using a hydraulic oil-filled remote microdrive (Drum drive; FHC, Bowdoin, ME, USA). Neurophysiological activity was amplified (variable gain; Grass P511 AC amplifier and high-impedance headstage), filtered from 0.03 to 3 kHz, and broadcast to the experimenters (outside the booth) over a monitor speaker. Action potentials were detected with a custom spike-discriminator circuit, consisting of a Schmitt trigger followed by a peak detector, and timed with National Instruments hardware. A wideband-noise search stimulus was used to isolate auditory-nerve fibers as the electrode was slowly advanced. Recordings were excluded as likely originating from nucleus angularis based on bipolar action potential morphology, non-primary-like post-stimulus time histograms in response to tones (i.e., onset or chopping responses), or weak phase locking to tones.

Tuning curves were determined through an automated algorithm that tracked the threshold sound level for rate excitation across frequencies. The algorithm estimated thresholds as the minimum tone level that repeatedly produced an increase in driven discharge rate compared to the most recent silent interval (*Chintanpalli and Heinz, 2007*). Tones were 50 ms in duration with 5 ms raised-cosine onset and offset ramps, presented once every 100 ms. Thresholds were sampled with a resolution of 28 frequencies per octave near characteristic frequency (CF), starting at 6–8 kHz and proceeding downward in frequency (typical range: 6 kHz to 200 Hz).

## SFOAE processing

Analyses were performed in MATLAB R2023a and R version 4.3.2. Occasional microphone-signal artifacts were rejected by thresholding. Surrounding each signal transient artifact, 20 ms of signal was removed. Responses to the probe, suppressor, and combined probe and suppressor were averaged

across repeated presentations. $P_{SFOAE}$ was calculated by vector subtraction (*Shera et al., 2002*), as indicated in *Equation 2*, where $P$ is the response waveform in Pa and subscripts denote the stimulus class.

$$P_{SFOAE} = P_{Probe} + P_{Suppressor} - P_{Probe+Suppressor} \qquad (2)$$

The noise floor was estimated by the spectrum of the difference between odd and even trials of SFOAEs. Magnitude and phase spectra were extracted from the mean data for each component of the signal using the generalized least-squares fit described in *Long et al., 2008*. Phase gradients were extracted after phase unwrapping and conversion to cycles. Two exclusion criteria were implemented prior to phase-gradient extraction, for $N_{SFOAE}$ estimation. Data points less than 10 dB in magnitude above the median of the estimated noise floor were rejected. Data points at the base of magnitude troughs were also rejected due to noted phase irregularities. Similar algorithms have previously been applied (e.g., *Shera and Bergevin, 2012*) to emphasize phase-gradient delays near magnitude maxima.

## Auditory-nerve response processing

Auditory-nerve tuning curves were first smoothed with a five-point triangular window. CF was calculated as the frequency of the lowest threshold, $Q_{10}$ was calculated as CF divided by the tuning curve bandwidth 10 dB above the lowest threshold, and $Q_{ERB}$ (i.e., the tuning quality of the idealized rectangular filter with equivalent bandwidth to the measured tuning curve) was calculated based on the area under the inverted tuning curve, as in *Bergevin et al., 2015*.

## Statistical analysis

Linear mixed-effects models were implemented to determine the effect of frequency on SFOAE magnitude and phase-gradient responses. Magnitude and $N_{SFOAE}$ values were grouped into five log-spaced frequency bands, with center frequencies ranging from 1 to 5 kHz. For each ear, first-degree robust polynomials were fit to the data in each frequency band. Magnitude and phase-gradient values were defined as the value of the polynomial fit at the center frequency in each band. Frequency was treated as a categorical variable. Ear intercepts were modeled as a random effect. The Satterthwaite approximation was used to calculate degrees of freedom for F tests. Statistical analyses were conducted with a significance level of $\alpha$=0.01.

## Acknowledgements

This research was supported by the National Institute on Deafness and Communication Disorders. Discussions with Natasha Mhatre are gratefully acknowledged.

## Additional information

### Funding

| Funder | Grant reference number | Author |
| --- | --- | --- |
| National Institute on Deafness and Other Communication Disorders | R01-DC017519 | Kenneth S Henry |
| National Institute on Deafness and Other Communication Disorders | R01-DC021953 | Kenneth S Henry |
| National Institute on Deafness and Other Communication Disorders | F31-DC021889 | Leslie Gonzales |
| National Institute on Deafness and Other Communication Disorders | R01-DC001641 | Laurel H Carney |

| Funder | Grant reference number | Author |
|---|---|---|

The funders had no role in study design, data collection and interpretation, or the decision to submit the work for publication.

## Author contributions

Diana M Karosas, Software, Formal analysis, Writing - original draft, Writing – review and editing; Leslie Gonzales, Yingxuan Wang, Formal analysis, Investigation, Writing – review and editing; Christopher Bergevin, Conceptualization, Writing – review and editing; Laurel H Carney, Conceptualization, Funding acquisition, Writing – review and editing; Kenneth S Henry, Conceptualization, Resources, Data curation, Software, Formal analysis, Supervision, Funding acquisition, Investigation, Visualization, Methodology, Project administration, Writing – review and editing

## Author ORCIDs

Christopher Bergevin ⓘ https://orcid.org/0000-0002-4529-399X
Kenneth S Henry ⓘ https://orcid.org/0000-0003-1364-318X

## Ethics

This study was performed in accordance with the recommendations in the Guide for the Care and Use of Laboratory Animals of the National Institutes of Health. All were handled according to a protocol approved by the University Committee on Animal Resources at the University of Rochester (UCAR-2011-015). All procedures were performed in anesthetized animals and every effort was made to minimize suffering.

Reviewer #1 (Public review): https://doi.org/10.7554/eLife.102911.3.sa1
Reviewer #2 (Public review): https://doi.org/10.7554/eLife.102911.3.sa2
Author response https://doi.org/10.7554/eLife.102911.3.sa3

# Additional files

## Supplementary files

MDAR checklist

## Data availability

The data used in this article and analysis code are freely available through the Open Science Framework at https://osf.io/qpuz5/.

The following dataset was generated:

| Author(s) | Year | Dataset title | Dataset URL | Database and Identifier |
|---|---|---|---|---|
| Henry KS | 2024 | Budgerigar otoacoustic emission study | https://osf.io/qpuz5/ | Open Science Framework, qpuz5 |

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
