## [Editor Report · eLife Assessment]

In contrast with mammals, measures of cochlear tuning in budgerigars do not match the frequency dependence of behavioral tuning. Earlier behavioral data in the budgerigar had shown good selectivity at around 3–4 kHz, but it was unknown whether this unusual selectivity arose in the inner ear or was a more central adaptation. The authors measured both auditory-nerve tuning curves and stimulus-frequency otoacoustic emissions and found fairly normal-looking cochlear tuning in the budgerigar. These **important** findings imply that any behavioral/perceptual differences in frequency selectivity are likely more central in original. These **solid** new data also provide significant support for the utility of otoacoustic estimates of cochlear tuning.

---

## [Referee Report · Reviewer #1 (Public review)]

Summary:

In their manuscript, the authors provide compelling evidence that stimulus-frequency otoacoustic emission (SFOAE) phase-gradient delays predict the sharpness (quality factors) of auditory-nerve-fiber (ANF) frequency tuning curves in budgerigars. In contrast with mammals, neither SFOAE- nor ANF-based measures of cochlear tuning match the frequency dependence of behavioral tuning in this species of parakeet. Although the reason for the discrepant behavioral results (taken from previous studies) remains unexplained, the present data provide significant and important support for the utility of otoacoustic estimates of cochlear tuning, a methodology previously explored only in mammals.

Strengths:

* The OAE and ANF data appear solid and believable. (The behavioral data are taken from previous studies and the resulting limitations are discussed.)

* No other study in birds (and only a single previous study in mammals) has combined behavioral, auditory-nerve, and otoacoustic estimates of cochlear tuning in a single species.

* SFOAE-based estimates of cochlear tuning were obtained by assuming that the tuning ratio estimated in chicken applies also to the budgerigar. Possible complications arising from an avian apical-basal transition analogous to that found in mammals are discussed.

---

## [Referee Report · Reviewer #2 (Public review)]

Summary:

Earlier behavioral data in the budgerigar have suggested frequency selectivity that was different from that in many other avian species, showing particularly good selectivity at around 3-4 kHz. It was unknown whether this unusual selectivity was determined in the inner ear, or whether it was a more central adaptation. The results using direct auditory-nerve tuning curves and less invasive stimulus-frequency otoacoustic emissions, suggest fairly normal-looking cochlear tuning in the budgerigar, implying that any behavioral/perceptual differences in frequency selectivity are likely more central in original.

Strengths:

- The study presents novel data in budgerigar, comparing the bandwidths of auditory-nerve tuning curves with the latencies of stimulus-frequency otoacoustic emissions (SFOAEs), which are thought to reflect the sharpness of cochlear tuning.

- Using a conversion factor taken from previous data in the chicken to avoid circularity of reasoning, the study shows quite good correspondence between the non-invasive estimates obtained from SFOAEs and the tuning obtained from auditory-nerve fibers. Similarity between budgerigar and chicken are harder to ascertain with the way the data are presented.

Weaknesses:

- The comparison of SFOAEs and auditory-nerve tuning curves in the most interesting regions (beyond 3.5 kHz, where some perceptual anomalies seem to occur in some previous data), relies on an extrapolation of the data from the chicken.

- No new behavioral data are presented, so the comparisons made in the paper are between studies separated by decades. None of the behavioral studies cited used the more current techniques that have been claimed to provide a behavioral estimate of cochlear tuning.

---

## [Author Response]

The following is the authors’ response to the original reviews

**eLife Assessment**
Previous studies in mammals and other vertebrates have shown that a noninvasive measure of cochlear tuning, based on the latency derived from stimulus-frequency otoacoustic emissions, provides a reasonable, and non-invasive, estimate of cochlear tuning. This valuable study confirms that finding in a new species, the budgerigar, and provides convincing support for the utility of otoacoustic estimates of cochlear tuning, a methodology previously explored primarily in mammals. The study's remaining claims of a mismatch between behavioral frequency selectivity and cochlear tuning are based on old behavioral data, and collected in an extreme frequency region at the edge of the limits of hearing. Hearing abilities are hard to measure accurately on the upper frequency edge of the hearing range, and the evidence for these claims is weak.

We appreciate the detailed summary of our paper by the editors highlighting its strengths. As described in the following responses, we added additional evidence to the Introduction supporting that budgerigars have (1) unusual behavioral frequency tuning compared to other bird species and (2) unusual behavioral tuning results in budgerigars are not readily explainable by the audiogram. This additional background information, including Fig. 1B, substantially strengthens the claim of mismatched behavioral and neural/otoacoustic frequency tuning in budgerigars. Moreover, that the behavioral data are “old” seems not particularly relevant considering that the same behavioral methods are still widely used in animal research, as elaborated upon in the responses below. We suggest the term “previously published” to clarify the behavioral data used in our analyses.

**Reviewer #1 (Public review):**
Summary:In their manuscript, the authors provide compelling evidence that stimulus-frequency otoacoustic emission (SFOAE) phase-gradient delays predict the sharpness (quality factors) of auditory-nerve-fiber (ANF) frequency tuning curves in budgerigars. In contrast with mammals, neither SFOAE- nor ANF-based measures of cochlear tuning match the frequency dependence of behavioral tuning in this species of parakeet. Although the reason for the discrepant behavioral results (taken from previous studies) remains unexplained, the present data provide significant and important support for the utility of otoacoustic estimates of cochlear tuning, a methodology previously explored only in mammals.Strengths:* The OAE and ANF data appear solid and believable. (The behavioral data are taken from previous studies.)* No other study in birds (and only a single previous study in mammals) has combined behavioral, auditory-nerve, and otoacoustic estimates of cochlear tuning in a single species.* SFOAE-based estimates of cochlear tuning now avoid possible circularity and were are obtained by assuming that the tuning ratio estimated in chicken applies also to the budgerigar.Weaknesses:* In mammals, accurate prediction of neural Q_ERB from otoacoustic N_SFOAE involves the application of species-invariance of the tuning ratio combined with an attempt to compensate for possible species differences in the location of the so-called apical-basal transition (for a review, see Shera & Charaziak, Cochlear frequency tuning and otoacoustic emissions. Cold Spring Harb Perspect Med 2019; 9:pii a033498. doi: 10.1101/cshperspect.a033498; in particular, the text near Eq. 2 and the value of CFa|b).Despite this history, the manuscript makes no mention of the apical-basal transition, its possible role in birds, or why it was ignored in the present analysis. As but one result, the comparative discussion of the tuning ratio (paragraph beginning on lines 383) is incomplete and potentially misleading. Although the paragraph highlights differences in the tuning ratio across groups, perhaps these differences simply reflect differences in the value of CFa|b. For example, if the cochlea of the budgerigar is assumed to be entirely "apical" in character (so that CFa|b is around 7-8 kHz), then the budgerigar tuning ratios appear to align remarkably well with those previously obtained in mammals (see Shera et al 2010, Fig 9).

We added sections on the apical-basal transition to the Results and Discussion, including how this concept might apply in budgerigars and other birds.

* For the most part, the authors take previous behavioral results in budgerigar at face value, attributing the discrepant behavioral results to hypothesized "central specializations for the processing of masked signals". But before going down this easy road, the manuscript would be stronger if the authors discussed potential issues that might affect the reliability of the previous behavioral literature. For example, the ANF data show that thresholds rise rapidly above about 5 kHz. Might the apparent broadening of the behavioral filters arise as a consequence of off-frequency listening due to the need to increase signal levels at these frequencies? Or perhaps there are other issues. Inquiring readers would appreciate an informed discussion.

This is a good point, also raised by reviewer 2, that declining audibility above 4 kHz could impact behavioral tuning estimates. On the other hand, other bird species with highly similar audiograms to budgerigars show conventional behavioral tuning that increases in sharpness relatively slowly and monotonically for higher frequences. Thus, the unusual pattern of behavioral tuning in budgerigars is not fully explainable by the audiogram. We added a section to the Introduction highlighting these points.

**Reviewer #2 (Public review):**
Summary:This manuscript describes two new sets of data involving budgerigar hearing: (1) auditory-nerve tuning curves (ANTCs), which are considered the 'gold standard' measure of cochlear tuning, and (2) stimulus-frequency otoacoustic emissions (SFOAEs), which are a more indirect measure (requiring some assumptions and transformations to infer cochlear tuning) but which are non-invasive, making them easier to obtain and suitable for use in all species, including humans. By using a tuning ratio (relating ANTC bandwidths and SFOAE delay) derived from another bird species (chicken), the authors show that the tuning estimates from the two methods are in reasonable agreement with each other over the range of hearing tested (280 Hz to 5.65 kHz for the ANTCs), and both show a slow monotonic increase in cochlear tuning quality over that range, as expected. These new results are then compared with (much) older existing behavioral estimates of frequency selectivity in the same species.Strengths:This topic is of interest, because there are some indications from the older behavioral literature that budgerigars have a region of best tuning, which the current authors refer to as an 'acoustic fovea', at around 4 kHz, but that beyond 5 kHz the tuning degrades. Earlier work has speculated that the source could be cochlear or higher (e.g., Okanoya and Dooling, 1987). The current study appears to rule out a cochlear source to this phenomenon.Weaknesses:The conclusions are rendered questionable by two major problems.The first problem is that the study does not provide new behavioral data, but instead relies on decades-old estimates that used techniques dating back to the 1970s, which have been found to be flawed in various ways. The behavioral techniques that have been developed more recently in the human psychophysical literature have avoided these well-documented confounds, such as nonlinear suppression effects (e.g., Houtgast, https://doi.org/10.1121/1.1913048; Shannon, https://doi.org/10.1121/1.381007; Moore, https://doi.org/10.1121/1.381752), perceptual confusion between pure-tone maskers and targets (e.g., Neff, https://doi.org/10.1121/1.393678), beats and distortion products produced by interactions between simultaneous maskers and targets (e.g., Patterson, https://doi.org/10.1121/1.380914), unjustified assumptions and empirical difficulties associated with critical band and critical ratio measures (Patterson, https://doi.org/10.1121/1.380914), and 'off-frequency listening' phenomena (O'Loughlin and Moore, https://doi.org/10.1121/1.385691). More recent studies, tailored to mimic to the extent possible the techniques used in ANTCs, have provided reasonably accurate estimates of cochlear tuning, as measured with ANTCs and SFOAEs (Shera et al., 2003, 2010; Sumner et al., 2010). No such measures yet exist in budgerigars, and this study does not provide any. So the study fails to provide valid behavioral data to support the claims made.

We appreciate the reviewer’s efforts in summarizing and critiquing our study. We feel that the budgerigar data collected by the Dooling and Saunders labs remain essentially valid today. The methods used in these behavioral studies are rigorous and remain widely used in animal research (e.g., critical bands and ratios: Yost & Shofner, 2009; King et al., 2015; simultaneous masking: Burton et al., 2018). The methods are based on the same power-spectrum-model assumptions of auditory masking as even the most recent and elaborate human psychophysical procedures. We therefore believe that it remains highly relevant to test and report whether these methods can accurately predict cochlear tuning. More importantly, while forward-masking behavioral results are hypothesized to more accurately predict cochlear tuning humans (Shera et al., 2002; Joris et al., 2011; Sumner et al., 2018), evidence from nonhumans is controversial. For example, one study showed a closer match between forward-masking results and auditory-nerve tuning (ferret: Sumner et al., 2018), whereas several others showed a close match for simultaneous masking results (e.g., guinea pig, chinchilla, macaque; reviewed by Ruggero & Temchin, 2005; see Joris et al., 2011 for macaque auditory-nerve tuning). Moreover, forward- and simultaneous-masking results can often be equated with a simple scaling factor (e.g., Sumner et al., 2018). Given no consensus on an optimal behavioral method, and seemingly limited potential for the “wrong” method to fundamentally transform the shape of the behavioral tuning quality function, it seems reasonable to accept previously published behavioral tuning estimates as valid while also discussing limitations and remaining open to alternative interpretations. We added these points to the discussion and added clarification throughout as to the specific behavioral approaches used.

The second, and more critical, problem can be observed by considering the frequencies at which the old behavioral data indicate a worsening of tuning. From the summary shown in the present Fig. 2, the conclusion that behavioral frequency selectivity worsens again at higher frequencies is based on four data points, all with probe frequencies between 5 and 6 kHz. Comparing this frequency range with the absolute thresholds shown in Fig. 3 (as well as from older budgerigar data) shows it to be on the steep upper edge of the hearing range. Thus, we are dealing not so much with a fovea as the point where hearing starts to end. The point that anomalous tuning measures are found at the edge of hearing in the budgerigar has been made before: Saunders et al. (1978) state in the last sentence of their paper that "the size of the CB rapidly increases above 4.0 kHz and this may be related to the fact that the behavioral audibility curve, above 4.0 kHz, loses sensitivity at the rate of 55 dB per octave."Hearing abilities are hard to measure accurately on the upper frequency edge of the hearing range, in humans as well as in other species. The few attempts to measure human frequency selectivity at that upper edge have resulted in quite messy data and unclear conclusions (e.g., Buus et al., 1986, https://doi.org/10.1007/978-1-4613-2247-4_37). Indeed, the only study to my knowledge to have systematically tested human frequency selectivity in the extended high frequency range (> 12 kHz) seems to suggest a substantial broadening, relative to the earlier estimates at lower frequencies, by as much as a factor of 2 in some individuals (Yasin and Plack, 2005; https://doi.org/10.1121/1.2035594) - in other words by a similar amount as suggested by the budgerigar data. The possible divergence of different measures at the extreme end of hearing could be due to any number of factors that are hard to control and calibrate, given the steep rate of threshold change, leading to uncontrolled off-frequency listening potential, the higher sound levels needed to exceed threshold, as well as contributions from middle-ear filtering. As a side note, in the original ANTC data presented in this study, there are actually very few tuning curves at or above 5 kHz, which are the ones critical to the argument being forwarded here. To my eye, all the estimates above 5 kHz in Fig. 3 fall below the trend line, potentially also in line with poorer selectivity going along with poorer sensitivity as hearing disappears beyond 6 kHz.

This is an excellent point, also raised by reviewer 1, that declining audibility above 4 kHz could influence behavioral tuning measures. While we acknowledge this possibility, declining audibility cannot fully explain the unusual pattern of behavioral frequency tuning in budgerigars considering that other bird species with the same audiogram phenotype show conventional tuning patterns. We added these points to the Introduction and Fig. 1B. We also added clarification throughout that it is not just the shape of tuning function that is noteworthy in budgerigars, but also the extreme slope in the 1-3.5 kHz region. Behavioral tuning quality in budgerigars increases by 5.3 dB/octave in this range (i.e., nearly doubling each octave increase in frequency), vs. 1.8 dB/octave in humans, 2.5 dB/octave in ferret, 1.1 dB/octave in macaque, and 1.9 dB/octave in starling. This additional background information, including Fig. 1B, substantially strengthens the claim of mismatched behavioral and neural/otoacoustic frequency tuning in budgerigars.

The basic question posed in the current study title and abstract seems a little convoluted (why would you expect a behavioral measure to reflect cochlear mechanics more accurately than a cochlear-based emissions measure?). A more intuitive (and likely more interesting) way of framing the question would be "What is the neural/mechanical source of a behaviorally observed acoustic fovea?" Unfortunately, this question does not lend itself to being answered in the budgerigar, as that 'fovea' turns out to be just the turning point at the end of the hearing range. There is probably a reason why no other study has referred to this as an acoustic fovea in the budgerigar.Overall, a safe interpretation of the data is that hearing starts to change (and becomes harder to measure) at the very upper frequency edge, and not just in budgerigars. Thus, it is difficult to draw any clear conclusions from the current work, other than that the relations between ANTC and SFOAEs estimates of tuning are consistent in budgerigar, as they are in most (all?) other species that have been tested so far.

We removed the term fovea from the paper. See above for our argument that unusual behavioral tuning in budgerigars is not simply or fully explainable by the audiogram.

**Recommendations for the authors:**

**Reviewer #2 (Recommendations for the authors):**
Line 34. As far as I could tell, no other study has referred to this region in budgerigar as an acoustic fovea. Probably for good reason (see above). This wording should probably be avoided.

We removed the term.

Line 35. Describing 3.5-4 kHz as 'mid-frequencies' is a stretch. 4 kHz is actually the corner frequency, above which hearing degrades.

We added a more detailed and accurate description of the tuning pattern.

Lines 89-91. This seems a nice statement of the problem, and to my mind makes for a much better rationale for the study.Line 255. "mixed effect" should "mixed effects".

We made the correction.

Line 380. Kuhn and Saunders didn't measure high enough to detect any changes in tuning.

We removed the reference here.